# Looking Back at the Early Stages of Redox Biology

**DOI:** 10.3390/antiox9121254

**Published:** 2020-12-09

**Authors:** Leopold Flohé

**Affiliations:** 1Dipartimento di Medicina Molecolare, Università degli Studi di Padova, v.le G. Colombo 3, 35121 Padova, Italy; l.flohe@t-online.de; 2Departamento de Bioquímica, Universidad de la República, Avda. General Flores 2125, 11800 Montevideo, Uruguay

**Keywords:** ferroptosis, glutathione peroxidases, heme peroxidases, hydrogen peroxide, lipid peroxidation, nitrogen monoxide radical, superoxide dismutase, superoxide radical, thioredoxin

## Abstract

The beginnings of redox biology are recalled with special emphasis on formation, metabolism and function of reactive oxygen and nitrogen species in mammalian systems. The review covers the early history of heme peroxidases and the metabolism of hydrogen peroxide, the discovery of selenium as integral part of glutathione peroxidases, which expanded the scope of the field to other hydroperoxides including lipid hydroperoxides, the discovery of superoxide dismutases and superoxide radicals in biological systems and their role in host defense, tissue damage, metabolic regulation and signaling, the identification of the endothelial-derived relaxing factor as the nitrogen monoxide radical (more commonly named nitric oxide) and its physiological and pathological implications. The article highlights the perception of hydrogen peroxide and other hydroperoxides as signaling molecules, which marks the beginning of the flourishing fields of redox regulation and redox signaling. Final comments describe the development of the redox language. In the 18th and 19th century, it was highly individualized and hard to translate into modern terminology. In the 20th century, the redox language co-developed with the chemical terminology and became clearer. More recently, the introduction and inflationary use of poorly defined terms has unfortunately impaired the understanding of redox events in biological systems.

## 1. Introduction

Life science means redox science. Even seemingly unrelated issues such as the hydrolysis of peptide bonds in living systems depend on enzymes, which have to be synthesized with significant consumption of ATP derived from redox processes in mitochondria or elsewhere. Thus, the attempt to review the evolution of the redox biology could easily result in a big historical volume looking back at all fields of biosciences. I am quite sure that this was not the idea behind the guest editor’s kind invitation to write this article, and I therefore take the liberty to narrow down the scope of the review. It will only cover the biochemistry of compounds called “ROS” or “RNS” for reactive oxygen or nitrogen species, respectively. I will try to describe the history of discoveries related to their formation in biological systems, preferentially in mammalian tissue, their pathophysiological and biological relevance and their metabolic fate. The key players of the redox events, the appreciation of their biological role and the language changed dramatically over the past three centuries and we may look forward to new surprises.

## 2. The Hydrogen Peroxide/Heme Period

Hydrogen peroxide (H_2_O_2_) was prepared from barium peroxide by the French chemist Louis Jaques Thénard (1777–1857) in 1818 [1], but the compound had to wait more than a century, before it found its place in biology. In fact, the discovery of enzymes degrading H_2_O_2_ preceded its unambiguous determination in biological systems. In 1863 the German/Swiss chemist Christian Friedrich Schönbein (1799–1868) reported that a large variety of plant and animal tissues could turn colorless guajacol tinctures into blue ones by hydrogen peroxide [2], and as shown in Figure 1, he classified this phenomenon as a catalytic oxidation. He therefore is commonly considered the discoverer of peroxidases (EC 1.11.1. …).

In the very same publication, Schönbein describes that these tissues, like platinum at high temperature, also liberate a gas upon H_2_O_2_ exposure that he characterized as “normal” oxygen. He also mentions that already Thénard had made similar observations with blood components. This means we could equally celebrate Schönbein or even Thenard as the discoverers of catalase (EC 1.11.1.6). Both catalytic activities, the catalatic and the peroxidatic one, proved to be heat-sensitive. Schönbein also mentions that a peroxidatic reaction had previously been seen in yeast by Julius Eugen Schlossberger (1819–1860).

Catalase, a peroxidase preferentially using H_2_O_2_ as a reductant for H_2_O_2_, in other terms a H_2_O_2_ dismutase, was then clearly established as an enzymatic entity of its own by Oscar Loew (1844–1941) in 1900 [3]. The elucidation of the structure and the reaction mechanism of catalase was greatly favored by the research on the cytochromes, which had already started in the 1920s [4,5]. According to Zámocký and Koller [6], it was Otto Warburg (1883–1970) who proposed iron catalysis as mechanistic principle of the catalase reaction in 1923. In the 1930s, Karl Zeile and colleagues noticed the similarity of the catalase spectrum and that of hemoglobin [7,8]. Finally, catalase was crystallized [9] and extensively investigated spectroscopically by Kurt Stern [10,11,12]. The spectroscopic investigations [10] clearly identified the prosthetic group of catalase as the same protoporphyrin IX present in the oxygen transporters hemoglobin and myoglobin and in most of the plant heme peroxidases. In his short Nature communication [11], he acknowledges the supply of reference compounds by Otto Warburg and Hans Fischer (1881–1950), which demonstrates the close cooperation of the groups working in the porphyrin field.

In case of catalase, the iron is coordinated to the four nitrogen atoms of the porphyrin ring, the fifth coordination site is bound to a protein histidine, while the sixth one interacts with the oxygen of water (ground state), H_2_O_2_ (first labile complex) or oxygen (compound I; see below). In ground-state catalase and peroxidases, the iron is usually in the ferric state. Extensive mechanistic, kinetic, electron spin resonance, sophisticated spectrophotometric and stop-flow techniques on catalase were later performed at the Johnson Foundation by Britton Chance (1913–2010) and others [13,14]. These investigations also led to the characterization of various catalytic intermediates. While still working in Stockholm with Hugo Theorell (1903–1982) on horseradish peroxidase (HRP) and catalase, Chance considered compound I as a typical Michaelis complex of the enzyme with H_2_O_2_ [15,16]. After having characterized compound I as a ferryl iron porphyrin IX-containing enzyme in both, peroxidases and catalase, it became clear that these oxidized enzyme forms are not complexes, but catalytic intermediates, i.e., covalently modified enzymes. With rate constants of <10^6^ M^−1^ s^−1^ (for both, the oxidizing and the reducing step, depending on species and condition), the catalase/H_2_O_2_ complexes have to be rated as extremely unstable, and more recent ab initio calculations have demonstrated that the ground-state catalase/H_2_O_2_ complex decays with a very low activation energy to generate compound I [17]. In the canonical catalytic cycle of catalase compound I, thus, is formed by oxidation of the ground-state enzyme by H_2_O_2_ and the ground-state enzyme is regenerated by reduction of compound I by H_2_O_2_.

In the early times, many attempts to detect H_2_O_2_ in living systems were made, but were mostly rated as unreliable because of a lack of specificity of the analytical tools. The difficulty to reliably detect H_2_O_2_ in situ did not only result from its low steady-state concentration (~10^−7^–10^−9^ M) but also from its rapid post mortem decline, which makes determinations in ex vivo samples practically impossible [18]. Stern [12], however, had already observed that the spectrum of catalase changed significantly upon H_2_O_2_ exposure. The oxidized catalase, which shows the distinct spectrum, is the catalytic intermediate that was later called compound I [14]. Stern therefore tried to detect intracellular H_2_O_2_ in situ by these spectral changes. He could, in fact, see the spectrum of oxidized catalase after having freed the rat liver from oxyhemoglobin by repeated washing with Ringer solution. Spectroscopic detection of compound I was later successfully applied by Chance to determine H_2_O_2_ production in a bacterial culture (*Micrococcus lysodeikticus*) [19]. More recently, when Christian de Duve (1917–2013) and colleagues had discovered the peroxisomes with densely packed catalase and various oxidases generating H_2_O_2_ in vitro, the formation of the catalase substrate in mammalian tissue could no longer be doubted [20,21]. Finally, Helmut Sies demonstrated compound I formation in hemoglobin-free perfused rat liver lobes [22]. Expectedly, the formation of compound I was particularly high, when substrates of peroxisomal oxidases such as octanoate, glycolate or urate were perfused [18]. The formation of H_2_O_2_ in cellular compartments other than peroxisomes required more sensitive analytical techniques. The scopoletin/HRP method developed by Andreae [23] in 1955 proved to be sensitive enough to unambiguously detect the H_2_O_2_ production at the zero-potential pool of the respiratory chain of pigeon heart mitochndria, which are apparently devoid of catalase [24]. In addition, the H_2_O_2_ adduct of yeast cytochrome *c* peroxidase has been successfully used as sensitive indicator of H_2_O_2_ generation, e.g., in the microsomal fraction [25].

Catalase has since been considered a major defense enzyme against H_2_O_2_, either by acting catalatically, as described, or peroxidatically, because the ground-state enzyme can also be regenerated from compound I by some low molecular weight reductants such as formic acid, methanol or ethanol [18]. The enzyme is rather specific for the oxidizing substrate; apart from H_2_O_2_ only some low molecular weight hydroperoxides without physiological relevance such as methyl or ethyl hydroperoxide are accepted [18]. Whether catalase acts catalatically or peroxidatically depends on the local concentration of reactants. A major site of catalase action in vivo is certainly the peroxisome, while other peroxidases dominate hydroperoxide metabolism in other subcellular compartments (see below; Section 3 and Section 4).

However, it is by no means justified to see the removal of H_2_O_2_ as a general in vivo function of peroxidases. The defense against hydroperoxide damage or the interruption of H_2_O_2_-mediated signaling is the job of catalase, of some members of the glutathione peroxidase family and the peroxiredoxins (see Section 4 and Section 6). For the majority of peroxidases, however, the biological role is still unclear. The best investigated example, the horseradish peroxidase, has found wide application in immune-histochemistry; its biological role, however, still remains obscure [26]. Very diverse functions of plant heme peroxidases have been implicated; they comprise fruit growth and ripening, signaling, seed germination, root and shoot elongation, auxin catabolism, defense response, wound healing, cell wall metabolism, lignification and suberization [26]. Of course, “ROS metabolism” is also listed [26]. However, it is difficult to imagine what all these diverse functions have in common with balancing hydrogen peroxide challenge. This diversity rather suggests that the typical plant heme peroxidases (class III peroxidases) make use of H_2_O_2_ for synthetic purposes or oxidation of specific targets. Intriguingly, a peroxidase of *Arabidopsis* appeared to enhance oxidative stress [27]. Similarly, the type III peroxidase Prx34 of the moss *Physcomitella patens* favors production of “ROS” via induction of NADPH oxidases and lipoxygenases [28]. In contrast, bacterial, fungal and plant class I peroxidases, which often also display catalase activity [29], and the ascorbate peroxidase of plants [26,30] and protozoa [31] appear to be involved in H_2_O_2_ detoxification, as evidenced by inverse genetics [32].

In the meantime, thousands of heme-containing peroxidases were detected, sequenced and, in part, structurally elucidated. It is as impossible as it is superfluous to describe the history of these heme peroxidases here in detail. This subject has been amply covered by countless reviews and monographs [17,26,33,34,35,36,37]. Herein, it may suffice to recall that in mammals, peroxidases can also serve synthetic purposes. A typical example is the thyroid peroxidase (EC1.11.1.8), which was discovered in the 1940s in Montevideo (Uruguay) [38] and has meanwhile been established as an enzyme responsible for the biosynthesis of the thyroid prohormone thyroxin [39,40,41]. Moreover, the peroxidases of white blood cells, the myeloperoxidase and the eosinophil peroxidase, and the lactoperoxidases are anything else but antioxidant enzymes. These peroxidases reduce H_2_O_2_ with halides or thiocyanate, thereby generating more aggressive oxidants such as hypohalous acids that may, e.g., chlorinate unsaturated lipids or phenolic compounds [42,43,44]. The products of the myeloperoxidases may generate singlet oxygen (usually abbreviated ^1^O_2_; mostly generated by light). ^1^O_2_ is an oxidant, for which no enzymatic detoxification is known. Its energy is, however, quenched by carotenoids (the rate constant for ^1^O_2_ quenching by ß-carotene is 1.1 x 10^10^ M^−1^ s^−1^ [45]; for review of older literature with focus on biological systems see [46]). Lactoperoxidase is therefore implicated in the prevention of bacterial infections in breast-fed children [47], and myeloperoxidase, first described as verdoperoxidase (EC 1.11.2.2) by Anger in 1941 [48], has meanwhile been established as a major antibacterial enzyme of polymorphonuclear leukocytes [49,50] and is equally considered to favor pathological processes presumably resulting from “oxidative stress” [51,52,53,54,55,56,57,58].

## 3. Selenium Conquering the Stage

Selenium was discovered 1817 or 1818 by Jöns Jacob Berzelius (Figure 2; the volume of *Journal für Chemie und Physik* is dated 1817, the Berzelius letter therein is from 1818) [59]. The element first made its career in diverse industries as chemical catalyst, semiconductor, in photocopying and coloring glass and ceramics [60]. In the biological context, it long remained an ugly smelling toxic, teratogenic and carcinogenic poison.

In veterinary medicine, selenium became known as the causative agent of diseases such as blind staggers of cattle and the “change hoof disease” of horses, when grazing on soils with high selenium content [61]. In the 1950s, however, Jane Pinsent observed that *Escherichia coli* required selenium for expressing optimum formic acid dehydrogenase activity [62], and Klaus Schwarz (1914–1978) found that rats deficient in vitamin E and selenium died from a fulminant liver necrosis (Figure 3) [63,64]. The real discoverer of the essentiality of selenium is hidden in an acknowledgement. It was the former deputy director of the NIH, DeWitt Stetten (1909–1990), who smelled the selenium in the lab of Schwarz (Thressa Stadtman (1920–2016), personal communication), while Schwarz had evidently developed a tachyphylaxis against the ugly smell of his selenium-containing factor 3 during the tedious isolation from hog kidney.

In the same year (1957), Gordon C. Mills described a peroxidase that was not affected by typical inhibitors of heme peroxidases such as azide or cyanide, proved to be highly specific for glutathione (GSH) and was claimed not to be a heme peroxidase [68]. This discovery was received with serious skepticism, and the new peroxidase, now known as glutathione peroxidase 1 (GPx1; EC 1.11.1.9), was declared not to exist at all at a Federation Meeting in the US by the father of peroxidase research Britton Chance (Gerald Cohen (1930–2001), personal communication). In consequence, less than one dozen of publications on GPx appeared within the 15 years after its discovery, mostly written by Europeans who could then not afford to cross the Atlantic for attending the Federation Meetings.

Alerted by an abstract claiming that glutathione peroxidase activity depended on selenium (Figure 3; [65]), we determined the selenium content in our crystalline preparation of bovine GPx1, which had survived 13 steps of purification, by neutron activation and found exactly 4 g atoms of selenium per mol of the homotetrameric enzyme [66]. As later reported by the groups of Albrecht Wendel and Al Tappel (1926–2017), the selenium in GPx is present as selenocysteine residue integrated in the peptide chain [69,70]. After confirmation of the selenoprotein nature, GPx became a celebrity and now entering the substance name “glutathione peroxidase” in EndNote yields >500 hits per year. Moreover, a large-scale preparation of GPx1 proved the absence of any known prosthetic group [71], and the bimolecular rate constant for the oxidation of GPx1 by H_2_O_2_ was faster (~5 × 10^7^ M^−1^ s^−1^, when calculated per subunit) than the corresponding one of catalase [67], which till then had been considered unbeatable in catalytic efficiency. Expectedly, the in situ function of GPx1 could be verified by a decrease in surface fluorescence due to NADPH consumption and release of oxidized glutathione in hemoglobin-free perfused rat liver, when H_2_O_2_ was not generated in the peroxisomes, but perfused [72,73,74].

We should mention here that not all members of the glutathione peroxidase family are selenoproteins. In four of the eight mammalian glutathione peroxidases the active-site selenocysteine residue may be replaced by cysteine and this appears generally the case in terrestric plants and bacteria (CysGPxs; Section 4.1). However, the second mammalian selenoprotein to be discovered also proved to be a glutathione peroxidase with an efficiency comparable to that of GPx1, the phospholipid hydroperoxide GSH peroxidase, now commonly named GPx4 (EC 1.11.1.12) [75]. These observations strengthened the belief that the magic catalytic power of selenium could substitute for the iron porphyrin prosthetic group of catalase and other heme peroxidases.

GPx4 proved to be distinct from GPx1 in sequence [76,77] and substrate specificity. While GPx1 accepts many soluble hydroperoxides apart from H_2_O_2_ [78], GPx4 also reduces the hydroperoxyl groups of complex, membrane-bound phospholipids [79]. In fact, GPx4 was discovered as an enzyme that prevented lipid peroxidation and was initially named PIP (for peroxidation inhibiting protein) [80], but later on, it proved to be the chameleon of the GPx family [81]. The *gpx4* gene is expressed in three different forms due to alternate use of start codons. The mitochondrial GPx4 is the one that forms the keratin-like matrix surrounding the mitochondrial helix in spermatozoa of mammals and, thus, is indispensable for male fertility [82]; the nuclear form has been implicated in chromatin compaction [83,84,85], while the cytosolic one primarily prevents lipid peroxidation by reducing lipid hydroperoxides and silencing of lipoxygenases [86].

Both, GPx1 and GPx4 have been reported to also efficiently reduce peroxynitrite to nitrite [87]. The rate constant for the peroxynitrite reduction by GPx1 is as high as 8 × 10^6^ M^−1^ s^−1^ [88]. Surprisingly, however, hepatocytes isolated from *gpx*^−*/*−^ mice proved to be resistant to a peroxynitrite challenge, one of the many paradoxical findings with antioxidant enzymes compiled by Lei et al. [89].

Intriguingly, the cytosolic expression form of GPx4 is the only member of the GPx family that proved to be essential [90], which highlights the importance of lipid peroxidation as a pathogenic principle [81]. In this context, it appears to be revealing that GPx4 has become known as the major antagonist of a special type of regulated cell death, ferroptosis. Ferroptosis is a multi-etiological phenomenon that is characterized by a common endpoint: an iron-catalyzed oxidative destruction of unsaturated lipids in bio-membranes. Screening for ferroptosis inhibitors inter alia yielded the compound SRL3, which irreversibly inhibited GPx4 [91] in the presence of 14-3-3ε as an auxiliary protein [92]. Therefore, ferroptosis was defined in 2015 as “iron-dependent form of regulated cell death under the control of glutathione peroxidase 4” [93]. In the meantime a lot of details of the ferroptotic process have been unraveled [94], and selenium, likely as an integral part of GPx4, has become an essential element to suppress ferroptosis [95]. It is therefore tempting to speculate that the fulminant liver necrosis that Schwarz saw in his selenium-deficient rats [64] was actually the first report on this peculiar form of cell death.

Studies on the mechanism of GPx started from the first x-ray analysis of bovine GPx1 by Rudi Ladenstein [96], who saw the selenium in close neighborhood of a tryptophan and a glutamine. The functional relevance of these residues was confirmed by Matilde Maiorino via site-directed mutagenesis [97]. The catalytic triad composed of selenocysteine, tryptophan and glutamine was later amended by an asparagine residue [98]. More recently, density functional theory (DFT) calculations [99] have demonstrated that the selenocysteine residue, after forming an adduct (or complex) with H_2_O_2_, is instantly oxidized without any activation energy to a selenenic acid derivative, which in two steps is reduced by thiols to regenerate the ground-state enzyme. The selenenic acid form could, however, not be verified by mass spectrometry, since the oxidized selenium reacts with a nitrogen of the peptide backbone in the absence of reducing substrate. In a homologous Cys-GPx (see below; chapter 4), however, the corresponding sulfenic acid derivative is clearly detectable. Splitting of the peroxide bond is achieved by a dual synchronized attack, a nucleophilic one by the dissociated selenol and an electrophilic one on the second oxygen by a proton bound in the active site, preferentially at the ring nitrogen of the tetrad tryptophan. These data perfectly match earlier kinetic studies on GPx1 displaying a ping-pong pattern (revealing an enzyme substitution mechanism) with infinite maximum velocity, infinite Michaelis constant and extraordinary efficiency [67,78].

Of course, selenium also plays a role in other redox processes. The 25 selenoprotein genes of the human genome overwhelmingly encode proteins that, according to their sequence, can be classified as oxidoreductases [100], the best investigated families being the thioredoxin reductases and the glutathione peroxidases. These two families of proteins unambiguously reduce hydroperoxides, either directly or indirectly via peroxiredoxins (see Section 4.2). The precise function of many selenoproteins is however still unknown. This even holds true for some of the glutathione peroxidases (for review see [101]). As to the other selenoproteins, we may refer to the dedicated monographs edited by Dolph Hatfield and colleagues [102,103,104,105]. Here, it may suffice to state that the myth of the magic catalytic power of selenium faded away sooner than anticipated.

## 4. Non-Se Glutathione Peroxidases, Peroxiredoxins and Other Super-Reactive Cysteine Residues

### 4.1. Cysteine-Containing Homologues of GPx

In order to demonstrate the catalytic importance of selenium, Rocher et al. exchanged the active-site selenocysteine residue in mouse GPx1 by its sulfur homologue cysteine and indeed saw a decline of three orders of magnitude in specific activity [106]. However, the specific activity, as measured under conventional test conditions (mM concentrations of both substrates), preferentially determines the rate of enzyme reduction by GSH [67] and, thus, provides limited information on the oxidation of the active site selenocysteine by H_2_O_2_. Similarly, the selenocysteine was replaced by cysteine in porcine GPx4 by Maiorino et al. [97]. In this investigation, both, the reductive and the oxidative part of the catalytic cycle were dramatically affected.

These findings were long considered to underscore the catalytic power of selenium versus sulfur. However, this view overlooks that the residual activities of the artificial Cys-GPxs are still orders of magnitude higher than any oxidation of a low molecular thiol by a hydroperoxide. Thiol oxidation by H_2_O_2_ was critically reviewed by Christine Winterbourn. The data compiled reveal that the rate constants of the reaction of any low molecular weight thiol compound with H_2_O_2_ never exceed 50 M^−1^ s^−1^, even if their thiol group is fully dissociated [107]. This sharply contrasts with the rate constants around 5 × 10^4^ M^−1^ s^−1^, as are seen in the artificial Cys-GPx4 [97]. Moreover, for the naturally occurring GPx of *Drosophila melanogaster,* which is a cysteine-containing GPx, the corresponding oxidative rate constant was determined to reach 10^6^ M^−1^ s^−1^ [108], thus coming close to those of its selenium-containing relatives. The seemingly poor GPx activity of many non-mammalian cysteine homologues of the GPx family results from a change of their substrate specificity; in functional terms, they are thioredoxin peroxidases [108,109]. Additionally, the DFT calculations mentioned above [99] showed that the mechanism of chalcogen oxidation by H_2_O_2_ is essentially identical for Sec-GPxs and Cys-GPxs. In short, the difference in catalysis of cysteine and selenocysteine residues in GPxs is not a qualitative, but only a quantitative one.

Just for completion, a non-selenium glutathione peroxidase, which showed preferential activity with organic hydroperoxides, was detected in liver of selenium-deficient rats by Lawrence and Burk in 1976 [110]. This enzyme proved not to belong to the GPx family, but is a B-type GSH-*S*-transferase [111].

### 4.2. Peroxiredoxins

High-to-extreme efficiencies in hydroperoxide reduction via sulfur catalysis is also observed in another thiol-dependent peroxidase family, the peroxiredoxins (Prx; EC 1.11.1.24–29). A peroxiredoxin was first seen in 1968 by Robin Harris in electron microscopy as a ring-shaped protein attached to erythrocyte ghosts [112]. It was not further investigated and, because of its peculiar shape, was called “torin” [113]. In the following years a considerable number of proteins that later turned out to be peroxiredoxins were described (for review see [114]). Their strange names disclose the lack of any serious functional characterization.

More defined examples of this family, the alkylhydroperoxide reductases, were detected in the group of Bruce Ames [115]. However, these researchers associated the reductase activity with the flavoprotein component (AhpF) of the system and thus overlooked the homology of the cofactor-free component (AhpC) with torin and other known sequences. In the late 1980s a “thiol-specific antioxidant protein (TSA)” had been isolated from *Saccharomyces cerevisiae* [116] and sequenced [117] in the early 1990s in the laboratory of Earl Stadtman (1919–2008). It proved to be a cofactor-free protein. I remember that Earl asked me at a meeting in Tutzing (Bavaria, Germany) to compare the then still unpublished sequence of TSA (1991) with that of GPx, because the GPxs were then the only known peroxidases just consisting of amino acids. The sequence did not show any similarity with that of any GPx. Instead, Chae et al. found out that the sequence of TSA was homologous to the AhpC component of the bacterial alkylhydroperoxide reductases [118] and a widely distributed family of proteins that later proved to be thioredoxin peroxidases [119,120]. When unsuccessfully chasing a GPx-homologous “trypanothione peroxidase” in the trypanosomatid *Crithidia fasciculata,* we stumbled across another peroxiredoxin. In the kinetoplasts, however, the specificity of the peroxiredoxin is a bit different; the kinetoplasts use the thioredoxin homologue tryparedoxin as reducing substrate [121,122,123].

A common denominator of the peroxiredoxin family proved to be a highly reactive conserved cysteine near the N-terminal end of the protein. Mutation of this cysteine results in inactivation in all peroxiredoxins so far investigated [119,124,125,126,127,128,129,130]; it was therefore named the “peroxidatic cysteine” (C_P_). The reactivity of this cysteine with hydroperoxides is facilitated by at least two more essential residues, an arginine [125,130] and a threonine, the latter being sometimes replaced by a serine [127,130]. Recent DFT calculations have unraveled that the oxidation of C_P_ is similar to that of the GPxs [131]. The proton dissociating from the C_P_-SH is kept in the reaction center, in case of the Prxs at the oxygen of threonine (or serine), while the arginine keeps the hydroperoxide in an optimum position by hydrogen bonding. Now the sulfur can start its nucleophilic attack on one oxygen of the peroxide bond, while the proton, unstably bound to threonine (serine) OH, combines with the other oxygen to generate water (or alcohol, depending on the substrate). As in the Cys-GPxs, the result of this initial step of the catalysis is an enzyme that has its C_P_ oxidized to a sulfenic acid, an intermediate detected long ago by Leslie Poole’s group [124].

The downstream reductive part of the catalytic cycle depends on the subfamily of Prxs. In subfamilies with a second conserved cysteine (“2-Cys-Prx”), the sulfenic acid of C_P_ forms an intermolecular disulfide bridge with the second, the resolving cysteine (C_R_), which is afterwards reduced by a redoxin (=protein characterized by an CxxC motif, typically thioredoxin; Figure 4). In the “atypical 2-Cys-Prxs”, the disulfide bridge is an intramolecular one. Their mechanism is, thus, analogous to that of Cys-GPxs with thioredoxin specificity. In the “typical 2-Cys Prxs”, which are the most common ones, two inter-subunit disulfide bridges are formed between two head-to-tail-oriented subunits [130,132]. In the “1-Cys-Prxs”, the reductive part of the catalytic cycle is mostly unclear. For human Prx6, GSH has been demonstrated to be the reducing substrate; but for GPx activity Prx6 requires GSH-*S*-transferase π as a supportive enzyme [133,134,135]. GSH dependence of a yeast 1-Cys-Prxs has also been discussed [136], but reduction of its sulfenic acid form by ascorbate has also been reported [137,138].

Like the GPxs, the Prxs have been implicated in the defense against a peroxide challenge. Both families reduce a broad spectrum of hydroperoxides. Some Prxs even reduce complex lipid hydroperoxides in vitro. Human Prx6, for instance, shares with GPx4 the ability to reduce phosphatidylcholine hydroperoxide [135]. Why Prx6 cannot substitute for the essential GPx4 in vivo is not known. In the meantime, Prxs have been detected in every domain of life and are increasingly discussed in the context of redox regulation (see Section 4.3 and Section 6).

### 4.3. Other Proteins with C_P_-Like Reactivity

Thiol groups of proteins that readily react with hydroperoxides are not restricted to the two thiol peroxidase families mentioned above. A typical example is the glycolytic enzyme glyceraldehyde-3-phosphate dehydrogenase (GAPDH), whose active site cysteine reacts with H_2_O_2_ with a bimolecular rate constant of 10^2^–10^3^ M^−1^ s^−1^ [132,141]. This oxidation is associated with a loss of glycolytic activity. As in GPxs and Prxs the reaction involves the primary formation of a sulfenic acid, as already shown by Little and O’Brien in 1969 [142], and results in numerous alternate functions that depend on the downstream reactions of the sulfenic acid form [143].

A second well investigated example is the transcription factor OxyR, which was discovered in the laboratory of Bruce Ames in 1985 as a regulon that responds to oxidative challenge in *Salmonella typhimurium* [144]. OxyRs are found in many bacteria, where they sense H_2_O_2_ and, upon oxidation, induce a large set of enzymes that inter alia protect against peroxide challenge. Their rate constants for the reaction with H_2_O_2_ range around 10^5^ M^−1^ s^−1^ [145]. Based on structural and kinetic data, Joris Messens and colleagues [145] proposed a mechanism that is reminiscent of those described for GPxs and Prxs. In addition, the thiol of a critical cysteine dissociates, its thiolate attacks the peroxide bond and becomes a sulfenic acid, while a proton kept in the reaction center combines with the second oxygen of the peroxide bond to create water as ideal leaving group. This dual attack on the peroxide, which is enabled by proton shuttling, has recently been confirmed in principle by DFT calculations [131].

Certainly, the examples here listed will not remain the only ones, and if one believes in redox proteomics, “reactive cysteine residues” are abundant in proteins. In fact, oxidatively modified cysteine residues (formation of inter- or intra-molecular disulfides, sulfenamides, persulfides, *S-*thiolated, or nitrosylated species) are easily detected and prevail in proteins involved in redox regulation. Oxidation of cysteine residues in the context of redox regulation or signaling is often called a “redox switch”, a term that, however, does not always consider the multitude of possible downstream-reactions and is used with different meanings. More importantly, the term “reactive cysteine” only makes sense, if the reaction partner is considered. Published rates for a direct cysteine oxidation by hydrogen peroxide in proteins are scarce and in most cases hardly comply with the assumption of a direct oxidation of such cysteines by H_2_O_2_ or any other hydroperoxide. This discrepancy of proteomic findings and kinetic data has been addressed in many reviews [107,146,147,148,149]. In recent years, new perspectives have helped to solve the enigma. The cysteine oxidation seen in ex vivo samples by protein chemistry indeed disclose a reactivity of these cysteines. They may be poorly reactive towards H_2_O_2_, but they react fast with oxidized thiol peroxidases; thereby, not only the kinetic barrier is overcome. Via specific protein/protein interaction, the unspecific oxidant H_2_O_2_ adopts specificity and, thus, becomes an ideal messenger in regulatory processes.

The first examples of this principle showed up in studies on redox regulation in yeast. The oxidative activation of the transcription factor Yap1 is here achieved by a thioredoxin-specific 2-Cys-GPx in *Saccharomyces cerevisiae* [150]. In *Schizosaccharamyces pombe* Yap1 activation by H_2_O_2_ is mediated by the 2-Cys-Prx Tsa1 [151,152]. In mammalian systems, Prx2 oxidizes the activator protein STAT3, GPx7 (and likely GPx8) uses protein isomerases as preferred reducing substrates [153] and, thus, contribute to oxidative protein folding in the endoplasmic reticulum [154,155]. GPx7 also oxidizes the glucose-regulated protein GRP78 (also called mortalin) and thereby improves its chaperone activity [156].

More recently, the group of Tobias Dick in Heidelberg (Germany) has demonstrated that 2-Cys-Prxs in mammalian cells facilitate the oxidation of regulatory target proteins. Knock-out of the cytosolic Prxs decreased the formation of disulfide bonds in cytosolic proteins [157]. This is exactly the opposite of what would have been predicted if the peroxiredoxins just competed for H_2_O_2_, and this clearly supports the idea that thiol peroxidases often act as sensors for hydroperoxides and, oxidized by the sensing process, hand over the redox equivalents to regulatory proteins [158].

## 5. The Biological Radicals

In the 18th and 19th century, the term radical indicated any group or substituent such as ethyl or carboxyl that was attached to a larger molecule [159]. The use of this term changed gradually, after Moses Gomberg synthetized a free and persistent radical for the first time, the triphenylmethyl [160]. Now, the term radical is restricted to compounds harboring one or more unpaired electrons and, in consequence, are paramagnetic. Compounds meeting these criteria are by no means uncommon in nature. In particular, enzymes or other proteins containing transition metals are often paramagnetic but are usually not named radicals.

Sometimes an unpaired electron resides in amino acid residues of the protein and is involved in the catalytic mechanism. The prototypes of the latter enzymes are the ribonucleotide reductases, which had been discovered in 1960 and the following years by Peter Reichard (1925–2018) and colleagues [161,162]. In 1972, Ehrenberg and Reichard provided the first evidence that the enzyme of *Eschericha coli* contained a free radical [163]. In 1978, finally the radical was identified as tyrosyl radical by electron spin resonance technology [164]. Depending on species and/or culture condition, the types of ribonucleotide reductases differ, but all make use of radical chemistry to eliminate the 2´-OH group of ribose in the ribonucleotide. In class Ia and Ib, an Fe-O-Fe bridge-stabilized tyrosyl radical attacks the ribose via a cysteyl radical, in class II the cysteyl radical is formed with the aid of adenosylcobalamine and class III works with a glycyl radical. The typical reductant of the ribonucleotide reductases is thioredoxin [165,166], glutaredoxin [167], other redoxins such as tryparedoxin [168] or formate (reviewed in [169,170,171]).

Another fairly stable free radical, ubisemiquinone, was detected in 1931 by Leonor Michaelis (1875–1949) [172]. In mitochondria, its oxidized and reduced forms are associated with complex I (NADH: ubiquinone oxidoreductase; EC 1.6.5.3) and complex II (succinate: coenzyme Q oxidoreductase; EC 1.3.5.1). They are, therefore, also called coenzyme Q, yet despite defined binding sites in the proteins of mitochondrial complexes, ubiquinone and ubiquinol can almost freely move within the mitochondrial membrane. The reduction of ubiquinone in complex I and II starts with a two-electron transition. In contrast, the cytochromes of complex III (coenzyme QH_2_: cytochrome c oxidoreductase; EC 1.10.3.2) and IV (cytochrome c oxidase; EC 1.9.3.1) transfer single electrons, which implies that somewhere in complex III or earlier a separation of electrons must take place, and ubisemiquinone would be a reasonable candidate to fulfill this job (but see below).

In the context of the present article, however, the focus should be on the really free radicals, i.e., those built by the organism on purpose, released from their site of generation and free to cause harm or benefit, wherever their life time allows them to diffuse. These are the superoxide radical anion (•O_2_^–^), its conjugate acid, the superoxide radical (•O_2_H), and the nitrogen monoxide radical (•NO; also called nitric oxide). The discovery of each of them came as an unanticipated surprise.

### 5.1. The Superoxide Radical

The superoxide radical was known to researchers interested in atmospheric chemistry or physico-chemists working with simplified clean systems [173]. As in the case of H_2_O_2_, the superoxide radical found its role in biology after its metabolism appeared at the horizon with the discovery of superoxide dismutase (SOD). The history of this discovery has been masterly reviewed by Irwin Fridovich (1929–2019). In the introductory chapter of the proceedings of the famous Banyuls symposium on “Superoxide and Superoxide Dismutases” (Banyuls, France; 1976), he amusingly describes the frustrated search for the explanation of a mysterious ferricytochrome *c* reduction that, strangely enough, depended on the presence of oxygen. The phenomenon had been observed in various biochemical reactions, the search for its chemical basis took decades, the methodologies became more and more complex, but no hypothesis could be experimentally verified. Finally, a youngster, Joe McCord, entered Fridovich’s lab, postulated that the reductant of cytochrome *c* could be superoxide, and identified SOD, which abolished the strange phenomenon, as an impurity in a carbonic anhydrase preparation they had used [174]. McCord’s hypothesis [175] indeed marks the beginning of superoxide research in biochemistry.

In 1969, superoxide dismutase was isolated from bovine erythrocytes [176]. It was the copper/zinc type that was known for years under different names for green proteins of unknown function such as hemocuprein, hepatocuprein [177], erythrocuprein [178] or cerebrocuprein [179]. The bimolecular rate constant for SOD-catalyzed dismutation of •O_2_^−^ is about 2 × 10^9^ M^−1^ s^−1^ [180] and, thus, is seven orders of magnitude faster than the non-catalyzed reaction (<100 M^−1^ s^−1^ [181]). The spontaneous dismutation at physiological pH is faster (~2 × 10^5^ M^−1^ s^−1^ [181]), since •O_2_^−^ is partially associated (pK_a_ = 4.8) and the dismutation of the protonated superoxide is faster (*k* for •O_2_H + •O_2_H = 7.6 × 10^5^ M^−1^ s^−1^ and for •O_2_H + •O_2_^−^
*k* = 8.5 × 10^7^ [181]). However, still SOD accelerates the dismutation by four orders of magnitude [181]. The rate constant of Cu/Zn-SOD is indeed the fastest ever reported for a bimolecular enzymatic reaction. The entire surface charge of the enzyme [182], and in particular an electrostatic gradient directed towards the reaction center guides the negatively charged superoxide radical anion towards the positive histidine-complexed copper ion [183,184], which explains the incredible efficiency of these enzymes.

In the following years, different types of superoxide dismutases were discovered: manganese- containing SODs in bacteria [185] and mitochondria of higher organisms [186], iron-containing SODs in bacteria [187] and protozoa [188] and extracellular forms of the Cu/Zn-SOD in mammals [189]. Cu/Zn-SODs were also sporadically found in bacteria. The first one was the enzyme of *Photobacterium leiognathi,* which lives as symbiont in the teleost pony fish. The unusual occurrence of a Cu/Zn-SOD in a symbiotic bacterium was suspected to be the result of a natural gene transfer [190]. However, sequencing of the Cu/Zn-SOD of *P. leiognathi* and comparison with known sequences falsified this assumption [191], and Cu/ Zn-SODs were soon discovered also in non-symbiotic bacteria [192].

As mentioned, the superoxide radical was discovered as a reductant, but it made its way in biology as an oxidant, since it can initiate and sustain free radical chains. With the availability of SODs, it became quite easy to prove the participation of superoxide in biological systems. The first pathogenic effect of superoxide formation was lipid peroxidation in biomembranes. As early as 1972, Fee and Teitelbaum described that oxidative hemolysis, as induced by dialuric acid, could be inhibited by SOD [193]. The basis of related experiments by Zimmermann and colleagues [194,195,196] were the rediscovery of catalase and glutathione peroxidase as contraction factor I and II by Albert Lehninger (1917–1986) and colleagues [197] and studies on high amplitude swelling of mitochondria induced by GSH [198,199]. These phenomena were shown to be associated with, and possibly caused by, lipid peroxidation in mitochondrial membranes. SOD indeed inhibited GSH-induced oxidative destruction of isolated mitochondrial membranes [196]. How the superoxide radical contributes to lipid peroxidation in this and similar artificial experimental settings, remains unclear. Certainly, GSH here does not act as an antioxidant; deprived of its enzymatic environment, it rather autoxidizes in the presence of traces of transition metals with formation of superoxide. Already in 1974, Misra had observed that autoxidizing thiols produce superoxide [200]. The superoxide radical (more likely than the superoxide radical anion) might abstract a hydrogen atom from a methylene group between two double bonds of a polyunsaturated fatty acid, which is the usual start of a free radical chain in membrane lipids. Accordingly, catalase and GPx1 inhibited loss of volume control and contractibility and lipid peroxidation [194,195,196,197].

These observations pointed to an essential contribution of H_2_O_2_ or any other hydroperoxide, respectively. A superoxide-driven formation of the hydroxylradical (•OH) from H_2_O_2_ in the presence of traces of iron, according to Haber and Weiss [173], might cause lipid peroxidation in simplified models such as washed mitochondria and isolated membranes. •OH is indeed a very aggressive oxidant. It reacts with a realm of naturally occurring compounds with rate constants higher than 10^9^ M^−1^ s^−1^, i.e., at rates near or at control by diffusion [201].

Strong oxidative power of H_2_O_2_ in the presence of Fe^2+^ had already been observed in the 19th century by the British chemist Henry J. Horstman Fenton (1854–1929) [202], but Fenton never mentioned the involvement of a radical, and the precise mechanism of the “Fenton chemistry” is still being debated. Most recently even a participation of singlet oxygen (^1^O_2_; the least excited species, ^1^Δ_g_O_2_, also occurs in biological systems) in such redox processes has been postulated [203]. This way, another oxidant would be added to the scenario of •O_2_^−^ products.

In short, even in simplified model systems of biomembrane destruction, we have to consider various initiators, propagators and amplifiers of free radical chains. Homolysis of H_2_O_2_ will yield two molecules of the hydroxyl radical, the most likely initiator of lipid peroxidation By analogy, homolysis of a fatty acid hydroperoxide would yield one hydroxyl radical and an alkoxyl radical (LO•), which implies that the radical chain would be accelerated due to branching. More Likely, however, •OH is generated from H_2_O_2_ or LOOH and Fe^++^ according to Haber and Weiss [173] or a Haber/Weiss-like reaction, respectively. In the latter case also an alkoxyl radical (LO•) may be formed, which is almost as aggressive as •OH [204]. After hydrogen abstraction (initiation), the polyunsaturated fatty acids usually add molecular dioxygen, which yields the lipid hydroperoxyl radical (LOO•). The latter can in turn abstract a hydrogen atom from another unsaturated fatty acid residue (propagation) or react with a chain-breaking scavenger such as vitamin E (termination). Singlet oxygen, as discussed in [195], is not involved, because spontaneous dismutation of •O_2_^−^ yields ground state oxygen [181]. In principle, however, also ^1^Δ_g_O_2_ may contribute to lipid peroxidation [205], if ^1^Δ_g_O_2_ is formed by myeloperoxidase products [206,207,208]. Apart from the canonical way of initiating lipid peroxidation, ^1^O_2_ tends to produce cyclic peroxides [205].

In vivo, lipid peroxidation is even more complicated. In mammals, up to eight lipoxygenases (COX and LOX) differing in reaction and substrate specificity contribute to lipid peroxidation (reviewed in [209,210,211]). They contain a non-heme iron and are usually dormant enzymes. Activation is achieved by oxidation of the catalytic iron, as has first been demonstrated for cyclooxygenase (COX1) in 1971 by William Lands and colleagues, and later extended to 5-LOX [212], 12-LOX [213] and 15-LOX [86]. Therefore, enzymatic lipid peroxidation is under the control of all enzyme families involved in hydroperoxide metabolism (reviewed in [211]), and some of the GPxs and Prxs also reduce the products of LOXs, the hydroperoxides, and, thus may act as terminators by preventing •OH formation from LOOH in a Haber/Weiss-like reaction. Of course, access to substrates has to be considered, which however remains unclear in many cases. Most of the thiol peroxidases require the support of a phospholipase, since, with the notable exception of GPx4, they can only reduce free fatty acid hydroperoxides efficiently, and the specificity for free fatty acids also holds true for most of the LOXs. Thus, biosynthesis and metabolism of lipid peroxides is under the control of lipases, in particular of phospholipase A_2_ and its regulator Ca^++^. The couple 15-LOX and GPx4 is an important exception, since 15-LOX appears unique in acting on complex phospholipids in membranes, thus producing the products that are specifically handled by GPx4 [81].

At the Tübingen (Germany) GSH meeting in early 1973, the role of •O_2_^−^ as a possible source of mitochondrial H_2_O_2_ was also discussed [214]. Gerriet Loschen and Angelo Azzi, who both attended and presented at the meeting, argued that the most likely source of the mitochondrial H_2_O_2_ was an autoxidizing cytochrome *b*, which, because of a maximum in the spectrum upon reduction at λ = 566, was called cytochrome *b*_566_. Since cytochromes typically catalyze single-electron transitions, not H_2_O_2_ but •O_2_H or •O_2_^−^ should be the primary product. It was already known that the mitochondrial matrix was filled up with MnSOD [185,215]. This enzyme could have instantly dismutated •O_2_^−^ and, in consequence, only H_2_O_2_ could have been detected outside of the mitochondria. Yet, the superoxide was not on Gerriet’s screen; Gerriet, now in Padova (Italy), kept fighting for his cytochrome *b*_566_ story with different arguments. So, I had to send another student of mine, Christoph Richter, to Padova, where Angelo had built up a small copy of the Johnston Foundation. Equipped with useful analytical reagents and Cu/Zn-SOD prepared in Tübingen [216] and thanks to Angelo’s support and instrumentation, Christoph could convincingly demonstrate the •O_2_^−^ formation in carefully washed inside-out mitochondrial particles [217,218]; and this finding was soon confirmed by others [219,220]. Later, using similar analytical tools, Richter could also show that some of the microsomal oxygenases primarily produce superoxide [221].

What followed was a fierce transatlantic debate about the precise mechanism of •O_2_^−^ formation in mitochondria. It was quite clear that it happened somewhere at the substrate site of the antimycin A block. Antimycin A blocks the respiratory chain at the oxygen site of cytochrome *b*_566_, which implies that all components at the substrate site of this block become reduced and can theoretically produce •O_2_^−^ by autoxidation. The problem is that there are so many components: the flavine of succinate dehydrogenase, non-heme iron proteins, ubiquinols and cytochrome *b*_566_. While Loschen and Azzi favored an autoxidation of cytochrome *b*_566_, the transatlantic party around Britton Chance, Alberto Boveris (1940–2020) and Enrique Cadenas insisted on autoxidation of the ubiquinols [222] and Boveris still appeared to defend this view in a recent review [223]. In 1986, however, Hans Nohl (1940–2010) and Werner Jordan reinvestigated the problem. They first showed that ubiquinol does not readily autoxidize and does not produce •O2^−^ in aprotic media such as mitochondrial membranes. Then, they made use of a novel inhibitor, myxothiazol [224], which had been isolated by Reichenbach and colleagues from *Myxococcus fulvus*. Myxothiazol blocks the respiratory chain at the substrate site of cytochrome *b*_566_ [225]. By means of this inhibitor, Nohl and Jordan could create a functional state of the respiratory chain with completely reduced ubiquinol and completely oxidized cytochrome b_566_. In contrast to antimycin A, myxothiazol did not induce any •O_2_^−^ production and antimycin A was no longer active in the presence of myxothiazol [226]. In particular the last quoted experiment unambiguously demonstrates that the mitochondrial •O_2_^−^ / H_2_O_2_ production, as detected by Loschen et al. [24,217], occurs in complex III, more precisely by autoxidation of cytochrome *b*_566_ and not by ubiquinol or the ubisemiquinone radical [214]. Yet by now, almost half a dozen different sites of mitochondrial superoxide production are being discussed, and the mechanisms differ [227,228]. An involvement of ubiquinols or flavin radicals can therefore not generally be ruled out.

An important beneficial role of •O_2_^−^ was reported in 1973. Bernhard Babior (1935–2004) et al. [229] demonstrated that granulocytes produced •O_2_^−^, and they already reasoned that this phenomenon was an essential part of the body’s defense system against pathogenic bacteria. The discovery was soon confirmed and extended to other phagocytes [230,231,232,233]. It complemented three fields of already advanced research: the respiratory burst known since 1933 [234], inflammation and phagocytosis known for more than a century by Elie Metchnikoff’s (1845–1916) milestone paper [235]. Already Metchnikoff had observed that phagocytosis was not only directed against bacteria, but the phagocytes attacked practically everything that is sick, dead or foreign, thus triggering an inflammatory response. Up to Babior’s discovery, H_2_O_2_ formed by the oxidative burst and halogen atoms (or hypohalous acids) arising from the myeloperoxidase reaction were widely considered the only bactericidal agents of phagocytosing leukocytes [236,237]. Initially, Babior appeared to believe that •O_2_^−^ itself was the predominant killing agent [229]. In the meantime we have learned that •O_2_^−^ is definitely the indispensable precursor of the H_2_O_2_ that is associated with phagocytosis, but the white blood cell use it also to make a highly toxic cocktail to cope with a bacterial invasion. It comprises •O_2_^−^, H_2_O_2_, •OH possibly derived from Haber/Weiss chemistry, •NO, peroxynitrite formed from •NO and •O_2_^−^ (see below), hypohalous acids or halogen atoms from a myeloperoxidase reaction, ^1^Δ_g_O_2_ and likely more, and the composition of the cocktail differs depending on the cell type (Figure 5). Moreover, oxidative burst and superoxide formation may occur independently from phagocytosis, if phagocytes are stimulated, e.g., by pro-inflammatory cytokines, immune complexes or the complement component C5a (compiled in [238]).

It appears needless to state that the bactericidal cocktail does not work without any collateral damage to the environment of a fighting leukocyte. It causes tissue damage and, in consequence, inflammation. Already before the superoxide dismutase became known, erythrocuprein was rediscovered as an anti-inflammatory protein under the name “orgotein”, which is in line with the pro-inflammatory role of •O_2_^−^ [241]. Orgotein was finally developed up to marketing approval in several countries for treatment of osteoarthritis, interstitial cystitis and *induration penis plastic*. Some years later, the drug had to be abandoned, because the promise of complete lack of antigenicity of the bovine protein turned out to be too optimistic. As a substitute, the recombinant human Cu/Zn-SOD was prepared in a hurry by Grünenthal GmbH (Aachen, Germany) and the Chiron corporation in Emeryville (CA; USA) [242,243] (Figure 6). The human SOD showed exciting promise in animal models of septicemia [244] or reperfusion injury [245], yet the general aversion against recombinant products in these years and the costs involved let the project die. In short, the hope for an improved clinical use of SOD [246] remained a dream.

Babior’s enzyme that produces superoxide radicals in phagocytes was first described by Sbarra and Karnowski in 1959, yet as an enzyme producing H_2_O_2_ [247]. It is now known as NADPH oxidase type 2 (NOX2) [248,249,250,251]. Its catalytic complex (p91^phox^ and p22^phox^) is a transmembrane protein. It contains an FAD and cytochrome *b*_558_ (discovered by Segal and Jones [252]). Its FAD moiety accepts the reduction equivalents of NADPH from the interior of the cell and releases •O_2_^−^ preferentially into the phagocytic vacuole, but also into the extracellular space (see below Figure 5). Like the lipoxygenases, NOX2 is a dormant enzyme that needs to be activated by cytosolic factors: p67^phos^, polyphosphorylated p47^phox^, p40^phox^, the GTPases Rac1 and Rac2, and Rap1. Any functional disturbance of this complex system leads to a severe clinical condition, chronic granulomatous disease, which is characterized by recurrent infections. The disease was first described in 1954 [253] and underscores the importance of NOX2 in host defense [240,254].

Superoxide production by NOX-type enzymes was soon detected also in many non-phagocytic cells. The sources are other members of the NOX family. The common denominator of these enzymes is a homologue of the flavocytochrome p91^phox^. However, their mode of activation and the pathologies in case of malfunction differ (compiled in [251]). In addition, not all NOX-type enzymes produce •O_2_^−^. DUOX I and DUOX II can make H_2_O_2_ directly and NOX4 appears to obligatorily produce H_2_O_2_ without the help of any SOD [251].

### 5.2. The Nitrogen Monoxide Radical

The discovery of the nitrogen monoxide (•NO; commonly called nitric oxide) did not only surprise, because it proved to be a radical—it also is a gas. The history has been reviewed by Salvador Moncada [255], Ferid Murad [256], Louis Ignarro [257], Robert Furchgott (1916–2009) [258,259] and Wilhelm Koppenol [260]. It started with the therapeutic use of nitro-vasodilators in the 19th century. A major push forward was the discovery of the endothelium-derived relaxing factor (EDRF) in the 1980s [261]. In many respects, EDRF mimicked the efficacy of compounds such as nitroglycerine or nitroprusside, but its chemical nature remained obscure. It was known that EDRF activated a guanylyl cyclase that had been extensively characterized by Murad [262], as did the nitro compounds, that it was inhibited by the superoxide anion radical, by hemoglobin and myoglobin and that it could be mimicked by •NO. In 1987, finally, two groups independently came to the very same conclusion: EDRF is •NO [263,264]. In 1998, Furchgott, Ignarro and Murad received the Nobel Prize “for their discoveries concerning nitric oxide as a signaling molecule in the cardiovascular system” [265]. Moncada and his colleagues, who published this discovery in the very same year [264] as Ignarro et al. [263], were not awarded. Only “The Nobel Assembly of Karolinska Institutet” could tell the reasons for this decision, but it did not [265].

Many of the physiological functions of •NO were already known around the time of the mentioned Nobel Prize [255,266]. Its target is a guanylyl cyclase, where it binds to a heme moiety and produces cGMP as the second messenger that leads to smooth muscle relaxation in practically all animals. In mammals its biosynthesis is achieved by three distinct nitric oxide synthases (NOS; nNOS, eNOS and iNOS for neuronal, endothelial and inducible NOS, respectively), which use L-arginine, NADPH and O_2_ as substrates and FAD, FMN, iron porphyrin IX, tetrahydrobiopterine and Zn ^2+^ as cofactors. Their functions differ. The essential function of eNos is the regulation of blood flow via production of EDRF; it also contributes to inhibition of platelet aggregation. The neuronal isozyme is involved in neurotransmission and synaptic plasticity. The inducible NOS is widely distributed, responds to exogenous stimuli such as bacterial lipopolysaccharides and phorbol esters and to endogenous pro-inflammatory cytokines. In macrophages, which typically lack myeloperoxidase, it complements the bactericidal cocktail with peroxynitrite, which is formed from •NO and •O_2_^–^ (see below). Apart from these canonical ways of •NO biosynthesis, the radical can also be produced by reduction of nitrite or nitrate [267].

In the meantime, •NO has reached the status of an approved drug to manage serious hypertension. A compound that inhibits the breakdown of its second messenger cGMP, sildenafil (Viagra^®^), has made its career as a lifestyle drug; it is used to improve penile erection. More recently, •NO was also discussed in plant and bacterial physiology. By mid July 2020, entering “nitric oxide” in EndNote yielded 88,863 hits, of which 10,853 were reviews. To discuss these more recent amendments in detail is simply impossible. We here can only touch upon some critical aspects.

•NO itself is a benign radical. Its biological effects are overwhelmingly beneficial. Its radical character, however, implies that it can react with a large variety of molecules and these down-stream processes may cause adverse effects. Fortunately, the history of nitrogen oxides can be traced back to Joseph Priestley (1733–1804), and a lot of the chemistry of •NO had been worked out before its presence in biological systems was detected [268]. The chemistry of the interaction of •NO with oxygen, thiols and other molecules is, however, very complex, and the relevance to biological systems still appears to be debated. Like •O_2_^–^, •NO can act as a reductant and as an oxidant.A prominent characteristic of •NO is its affinity to metal complexes. It is the basis of its physiological function as activator of guanylyl cyclase, but also of adverse effects resulting from binding to cytochrome P450 in the endoplasmic reticulum or to the cytochromes of the respiratory chain. The interaction of •NO with *b*-type cytochromes in complex III appeared to mimick antimycin A in triggering superoxide production (see above), which implies the formation of peroxynitrite (ONOO^−^) due to the simultaneous presence of •NO and •O_2_^–^ and, in consequence, mitochondrial dysfunction [269].•NO can interact with the biradical molecular dioxygen to form a realm of nitrogen oxidation products comprising radical and non-radical species such as, e.g., •NO_2_, •N_2_O_2_^–^, N_2_O, N_2_O_3_, NO^–^, NO_2_^−^, ONOO^−^ and NO_3_^−^ [268,270].In contrast to •NO, •NO_2_ is a strong oxidant and is likely responsible for nitration of tyrosine in proteins [271]. The bimolecular rate constant for the reaction of •NO_2_ with tyrosine at pH 7.5 is 3.2 × 10^6^ M^−1^ s^−1^, •NO_2_ will also nitrate unsaturated fatty acids [268,272].Nitrosothiol in proteins or low molecular compounds such as GSH is commonly considered as a hallmark of “nitrosative stress”. Of course, these derivatives could be formed by a reaction of •NO with thiyl radicals, yet the steady state concentration of thiyl radicals is too low to be of physiological relevance. Most likely, *S*-nitrosation is achieved by N_2_O_3_, the latter being built from •NO and •NO_2_, with a rate constant of 1.1 × 10^9^ M^−1^ s^−1^ [268]. However, also other mechanisms are being discussed [270].In the context of lipid peroxidation, •NO can adopt controversial roles. Being a radical, it can terminate free radical chains, e.g., by interacting with an ROO• [273]. Its oxidation products, however, may also initiate a free radical chain by hydrogen abstraction from a poly-unsaturated fatty acid residue [272].The most important pathogenic reaction of •NO is probably its combination with •O_2_^−^ to form peroxynitrite. This reaction of two radicals proceeds with a rate constant of 1.9 × 10^10^ M^−1^ s^−1^, which implies that it is limited by diffusion [270,274]. Peroxynitrite, although it is not a radical, is a highly aggressive oxidant, which prompted Beckmann and Koppenol to describe this reaction as one of the “good” (•NO) with the “bad” (•O_2_^−^) to make the “ugly” (peroxynitrite) [275].Peroxynitrite, apart from being detrimental by itself, had been proposed to decompose into NO^−^ and ^1^Δ_g_O_2_, thus creating another aggressive oxidant [276]. This hypothesis was, however, falsified by two later publications [277,278].•O_2_^−^, by reacting with •NO to peroxynitrite, inhibits the beneficial effects of •NO, e.g., on the circulation [279,280], and simultaneously causes oxidative damage. In retrospect, therefore, the surprising results seen with SOD infusion in models of reperfusion injury and septicemia [246] may be re-interpreted as resulting from •NO salvage and inhibition of the formation of peroxynitrite.

In short, •NO itself is beneficial in guaranteeing optimum blood flow and neuronal function, but when transformed to •NO_2_ or peroxynitrite, it becomes Janus-faced: it creates an efficient bactericidal cocktail with the typical collateral oxidative tissue damage [272,281,282,283]. For recent developments and ramifications in the field see [267,284,285,286].

## 6. Changing Paradigms: From Tissue Damage to Redox Regulation

The interest in free radical biochemistry was certainly boosted by Denham Harman’s (1916–2014) free radical theory of aging published in 1956 [287] and Rebeca Gershman’s (1903–1986) observation of the similarity of oxygen poisoning and x-ray irradiation damage [288]. However, our knowledge on the process of aging has since apparently not improved [289]. Nevertheless, the concern about oxidative damage of DNA, proteins and other macromolecules kept the redox biochemists busy for some decades. Indeed, all related symposia and monographs of the last century focused on oxidative tissue damage [239,290,291,292,293,294,295]. The recent revival of lipid peroxidation research in the context of ferroptosis reveals the importance of the past efforts [296,297]. With the beginning of the new millennium, the journals, meetings and monographs on redox biology are almost exclusively filled with contributions on redox regulation and redox signaling. Although flourishing quite late, the subject of redox-dependent regulatory phenomena is not new at all.

Early observations of a redox-dependent metabolic regulation were already reported in the 1960s. Jacob and Jandl [298] saw an activation of the pentose phosphate shunt upon oxidation of GSH in red blood cells. Pontremoli et al. [299] described that fructose 1,6-diphosphatase, a key enzyme in gluconeogenesis, was activated by incubation with cystamine and that this activation could be reversed by thiols such as GSH, cysteine or mercaptoethanol. The authors attributed this effect to an oxidative modification of a particular cysteine residue of the enzyme. In the 1970s, Czech et al. speculated on the involvement of SH oxidation in the action of insulin [300,301] and May and de Haen proposed that H_2_O_2_ might be a second messenger of insulin [302]. The hypothesis was finally corroborated by over-expression of GPx1; the mice became fat and insulin resistant [303]. By now, the role of H_2_O_2_ in insulin signaling is firmly established [304]. In 1974, Eggleston and Krebs further discussed a regulation of the pentose phosphate shunt by GSSG [305], and in 1985, Regina Brigelius could already compile a large list of proteins that were *S*-modified, mostly glutathionylated, under oxidant conditions, although the mode of their formation and the functional consequences were still unclear in many cases [306].

Despite these early hints suggesting a metabolic regulation by redox events, phosphorylation and de-phosphorylation dominated the field for quite a while (for a historical review see [307]). The delayed merging of this research field with redox biology must indeed surprise, since the oxidative inactivation of protein phosphatases had been known already in the 1970s [308]. The situation only changed when it became obvious that many phosphorylation cascades are under the control of oxidants, the insulin case mentioned above being just one of many examples. In particular, growth factor signaling proved to be redox-controlled (for reviews see [309,310,311,312]). Early examples are the oxidative activation of NFκB [313,314,315,316], the activation of a NOX by binding of transforming growth factor ß to its receptor [317], and NOX activation upon receptor occupancy by platelet-derived (PDGF) or epidermal growth factor (EGF) [318,319,320]. According to Sundaresan et al. [318], the synchronization between binding of EGF, PDGF, IL1, and NFκB to their receptors and the activation of NOX is mediated by the small GTP-binding protein RAC1, yet the mechanisms certainly differ between cellular compartments, cell types and phosphorylation cascades. More recently, even the participation of mitochondrial •O_2_^−^/H_2_O_2_ in redox regulation is being discussed [227].

In most cases, the regulating compound remains hidden in the cloudy abbreviation ROS (see Chapter 7). Bacteria, however, have distinct sensors for H_2_O_2_, •O_2_^−^ and lipid hydroperoxides (reviewed in [321]). H_2_O_2_ specifically oxidizes the transcription factor OxyR, already introduced in Section 4, the SoxR regulon appears to specifically respond to •O_2_^−^ [322,323], and Ohr, which is a peroxiredoxin, only reacts with lipid hydroperoxides [324]. These systems collectively induce an adaptive transcriptional response that makes the bacterium more resistant to oxidant challenges.

In mammals, sensing ROS usually means sensing H_2_O_2_, although more specific sensors may also exist in higher organisms. In fact, NF-κB activation, which has long been known to be favored by oxidant conditions [313,315], can be inhibited by overexpression of GPx1 [314], but interleukin-induced NF-κB activation was more efficiently inhibited by overexpression of GPx4, which points to a critical role of lipid hydroperoxides in the activation cascade [325]. Additionally, the novel form of regulated cell death, ferroptosis, is selectively affected by GPx4. As a general rule, the less reactive species such as H_2_O_2_, lipid hydroperoxides or the superoxide radical anion are more likely involved in regulation, while the aggressive oxidants such as the •OH radical, singlet oxygen or peroxynitrite react too promiscuously to allow a specific regulation and are more likely just damaging.

The known molecular targets of regulatory redox events are almost exclusively thiols in proteins. The latter may be oxidized directly by H_2_O_2_ or another hydroperoxide, as outlined already in Section 4. Alternatively, thiol-dependent peroxidases, which are particularly competent for the reaction with hydroperoxides, may serve as primary sensors and transfer their redox equivalents via heterodimer formation followed by disulfide reshuffling to the ultimate target proteins (for details see Section 4). The redox regulation of phosphorylation cascades can be achieved in different ways. Either a protein kinase can be activated by cysteine modification at the sequence motif MxxCW [326] or, more commonly, protein phosphatases may be inhibited by oxidation of their active-site cysteine. The net effect, inhibition or activation, depends on the role of respective kinases and phosphatases in the cascades. Since the majority of protein phosphatases are susceptible to oxidative inactivation, practically all phosphorylation cascades proved to be redox-regulated.

The adaptive response in mammals is predominantly regulated by the Nrf2/Keap1 system [146]. Nrf2 was detected by Itoh et al. in 1997 as a master regulator for a realm of cytoprotective enzymes [327]. It is kept in the cytosol by Keap1 and continuously degraded. Keap1 contains a reactive cysteine residue, the oxidation or alkylation of which allows Nrf2 to migrate to the nucleus, where it activates the transcription of ARE-regulated genes. ARE stands for “antioxidant response element”, which actually is a misnomer, since its activation depends on the oxidation of Keap1, as outlined above. The name ARE is therefore often replaced by the more adequate abbreviation EpRE for electrophile response element [328,329]. The original name ARE was chosen because its activation was achieved by synthetic antioxidant phenols [330]. Such compounds, however, tend to react with the most abundant radical of biological systems, which is the normal triplet oxygen, and thereby produce oxidants instead of scavenging dangerous radicals. This behavior of the synthetic phenols is shared by many antioxidants, also by natural ones, which implies that, in vivo, they act like oxidants. This way, however, they may induce the expression of protective enzyme via the Nrf2/Keap1/EpRE system and, thus, improve the antioxidant defense system. This appears to be the mechanism, by which antioxidants, if active at all, could display beneficial health effects [331,332]. They might trigger hormesis, a concept attributed to Theophrastus Bombastus von Hohenheim (“Paracelsus”; 1493–1541).

Other important regulators that work with reversible thiol oxidation are the redoxins. Typically, the reduced form binds to a transducer or adapter protein and thereby blocks signal transduction. Upon oxidation, the redoxin is released, and signaling can proceed. A classical example is the redox-sensitive association of thioredoxin with ASK1, the apoptosis signaling kinase [333]. Similarly, reduced nucleoredoxin (a redoxin with a CPPC motif) binds to the adapter protein MyD88 and thereby blocks recruiting of Myd88 to the Toll-like receptor [334], and reduced nucleoredoxin also interrupts Wnt signaling by redox-sensitive binding to Dvl (disheveled) [335].

Although the quoted examples demonstrate the importance of protein thiol modification in regulatory processes (for more details see [158,321,336,337,338,339], other mechanistic possibilities should not be ignored.

A reversible oxidation of the sulfur in methionine residues and its functional consequences have more recently been demonstrated [340]. The already mentioned oxidation of iron in lipoxygenases is likely independent of any sulfur modification. In principle, every redox-sensitive residue or component of a protein is susceptible to redox regulation. Moreover, sometimes an activity modification affects seemingly remote metabolic pathways [341]. A typical example is the oxidative inactivation of GAPDH activity (see Chapter 4), which blocks glycolysis and directs the glucose metabolism to the pentose phosphate shunt. We therefore should remain open to still undiscovered regulatory phenomena.

An almost topical state-of-the-art review and a compilation of ongoing research has recently been published as “a summary of findings and prospects for the future” by the ~100 participants of the COST Action CM1203 (“EU-ROS”) [342].

## 7. Now the Language Problem

The title of this Special Issue “Redox language of the cell” deserves a critical comment: Cells do not talk to us, we talk about cells. This means that the redox language is a human one and, as such, prone to errors and unfavorable developments.

Before the 20th century the language of (bio)chemists was very individualized and often hard to understand. Accordingly, it was not always easy to figure out, who invented what. For instance, Veitch in the short historical introduction of the horse radish peroxidase review [35] traces the discovery of plant peroxidases back to times before the discovery of H_2_O_2_. Louis Antoine Planche (1776–1840) reportedly described the development of blue color in “jalap” by fresh horse radish in 1810 [343]. Schönbein, who is most often quoted as the discoverer of peroxidases (see Section 2), essentially performed the same experiments, just with more tissue samples and (wrongly defined) H_2_O_2_. His language and his mechanistic speculations are strange, to say the least (see Figure 7). In his article [2], the blue color of guajacol (jalap) is also produced by ozone, “metal superoxydes” (or proxides), platinum and other noble metals in the presence of H_2_O_2_ (called “Wasserstoffsuperoxyd”, a name now reserved for the hydrogen superoxide radical and confusingly also presented with the composition of the latter). All these compounds like the catalysts of natural sources are believed to generate an activated form of oxygen that oxidizes the guajacol resin.

Sometimes Alexander von Humboldt (1769–1859) is named the founder of redox chemistry, because he is presumed to have described the production of barium peroxide, which Thénard used to prepare H_2_O_2_. I checked Humboldt’s pertinent publication [344] carefully, but was unable to find an unambiguous proof of this assumption; the description of the starting materials (“Alaun-Erden” or “schwere Erden”) were just too unprecise to understand what kind of chemical experiments he performed.

Over the 20th century, the language of redox biology co-developed with the chemical terminology. It was complemented by the UPAC enzyme nomenclature and some abbreviations that were commonly understood. By the third quarter of the century, the language of redox biologists had reached a maturity that allowed an easy communication between reasonably educated scientists of different disciplines. Unfortunately, this favorable development was discontinued. Beginning in the last quarter of the 20th century, the redox biologists enjoyed creating their own language—and created little else but confusion. Fragments like “free radicals collectively named ROS….”, “free radicals such as H_2_O_2_…”, “GPx scavenges free radicals…” or “transfers free radicals from intracellular ROS to glutathione” are not rare in the scientific literature (politeness forbids the referencing of such statements).

What follows below should not be interpreted as an undue outbreak of frustration; it is to express my concern about the growing lack of precision in more recent publications. This concern is shared by others. Also, the report on the COST Action CM1203 starts off with a harsh critique of the use of ill-defined terms [342], and Henry J. Forman, on behalf of the entire board of “*Free Radical Biology & Medicine*” published an article with the title: “Even free radicals should follow some rules: a guide to free radical research terminology and methodology” [345]. The latter publication discourages experimentation with methods, the specificity of which is defined by little else than the promotional material of the kit industry, and complains about the use of poorly defined “scientific” terms. Here, I will not reiterate all arguments of these publications. A few remarks should suffice to alert uncritical redoxologists.

In 1985, Helmut Sies promoted the term *“oxidative stress”* by using it as book title [291]. There, the term was introduced to describe pathological conditions in which the endogenous defense mechanism can no longer cope with the production of oxidants. Such situations do exist, e.g., in septicemia, ischemia/reperfusion and certain poisonings, and accordingly the term made a lot of sense. Over the years, the oxidative stress was repeatedly re-defined to consider new aspects of redox biology [346,347,348,349,350,351]. However, these modifications of the term were not always and not instantly accepted by the scientific community. In consequence, the precise meaning of oxidative stress varies with the date of publication and the authors. In the meantime, every tissue, cell or subcellular compartment had its own poorly defined form of stress. The oxidative stress has been divided into a “distress” and an “eustress” and complemented by a “reductive stress”. The redox-related stresses, thus, fused to a continuum that leaves no room whatsoever for unstressed normal life. In fact, physiological redox regulation, which is commonly not perceived as stressing, is now found under a broad stress umbrella as a form of eustress [351]. In short, the term stress has lost its power to differentiate between a pathological event, an adaptation to changing challenges and physiological fine-tuning of metabolic fluxes by redox regulation or redox signaling. This scenario justifies the question of whether the original definition of oxidative stress [291], which described a disturbance with pathological consequences, was not clearer.

Other examples of poorly defined terms are “*ROS*” and “*RNS*” for reactive oxygen and nitrogen species, respectively. They are often used as synonym for radical species, although some of the aggressive derivatives of oxygen and nitrogen are not radicals, and some of the radicals are not very reactive. It also has become fashionable to hybridize ROS and RNS to RONS or to invent subtypes of ROS such as mROS, if a mitochondrial origin is suspected, or lipid ROS, again without a clear definition. So far, I could not find out if the highly unstable thromboxane A_2_, the peroxides prostaglandin G_2_ and H_2_ or the epoxide leukotriene A_4_ belong to lipid ROS or not. Finally, the borderline between ROS and RNS is unclear. Most of the RNS contain more oxygen than nitrogen, and depending on the mesomeric form, a lone electron of an RNS may be centered at the oxygen. Admittedly, it is sometimes difficult to precisely state which species is responsible for a reaction. This is particularly true, when the bactericidal cocktails of phagocytes are involved. However, this difficulty should not be misused as an excuse, not to head for the clearest language possible. I feel heavily distressed, when seeing exploding stars spreading ROS all over a paper.

A related problem is hidden in the term “*reactive*”. It has already been mentioned (see Section 4) that “reactive” is meaningless without naming the reaction partner. Normal triplet oxygen (^3^O_2_) is never considered a reactive species, but it binds fast to hemoglobin and myoglobin and reacts fast with cytochrome *c* oxidase and a realm of oxygenases. H_2_O_2_ does not display any exciting rates with low molecular weight thiols (*k* ≤ 50 M^−1^ s^−1^ [107]), but it may react with protein thiols in GPxs and Prxs with rate constants higher than 10^6^ M^−1^ s^−1^ [352,353]. The superoxide radical anion hardly attacks amino acids. The highest rate constant was seen with tryptophan (24 M^−1^ s^−1^ [354]), but it reacts fast with hydrated copper (2.7 × 10^9^ M^−1^ s^−1^ [355], depending on conditions, up to 8.1 x 10^9^ M^−1^ s^−1^ [182]), with the copper of Cu/Zn SOD (2 × 10^9^ M^−1^ s^−1^ [181]) and with •NO (1.9 × 10^10^ M^−1^ s^−1^ [270]).

The most serious confusion came from the term *antioxidants.* When I was still a student, I learned from chemical textbooks [356,357] that antioxidants were used in polymer chemistry. The term described chain terminators in peroxide-initiated radical polymerization of olefins. According to Di Meo and Venditti [321], the term with exactly this meaning was introduced by Moureau and Dufraisse [358] already in 1926. Later, the antioxidants were hijacked by biochemists to describe inhibitors of analogous free radical chain reactions in living organisms. Meanwhile this term is degenerated to describe something that is presumed to be good for human health. The idea that this perception is not simply biased may be exemplified by glucose, which is thought not to be good for human health and is (perhaps as a consequence) never called an antioxidant. As a polyol, however, glucose is an efficient free radical trap (*k* for the reaction with •OH =1 × 10^10^ M^−1^ s^−1^ [201]) and, when metabolized, is the main source of reduced pyridine nucleotides, the small currency of redox biology, which guarantees defense against peroxide challenge and repair of oxidant damage. A clear definition of antioxidants no longer exists. They comprise free radical trapping agents such as polyphenols, substrates of enzymes (e.g., GSH), which by themselves are poor or no antioxidants at all, and compounds that become constituents or cofactors of enzymes (e.g., selenium or zinc).

Intriguingly, despite the heavy promotion of antioxidants in the lay press, controlled clinical trials in various “oxidative stress diseases” yielded overwhelmingly negative results (no or adverse effects) [89,342]. The reasons for these seemingly paradoxical observations are likely trivial. First, the steadily growing list of oxidative stress diseases is not defined either. The expression rarely distinguishes between a pathogenic relevant oxidant challenge and clinically irrelevant increase in •O_2_^−^/H_2_O_2_ production, as is seen in almost every disease. Second, the most aggressive and damaging radical, •OH, reacts fast with almost everything, with thiols, polyols, aromatic compounds including phenols, purines and pyrimidines with rate constants beyond 10^9^ M^−1^ s^−1^ [201]. The concentration of such endogenous targets of the •OH radical may be estimated to lie in the medium to high millimolar range; a most efficient radical-trapping antioxidant would have to reach as similar concentration in vivo in order to significantly reduce an •OH-induced tissue damage. This is unfortunately not possible without drastically increasing the volume of the patient. This consideration, incidentally, may also explain why nature has not developed any device to cope with an •OH challenge. The enzymatic systems to fight oxidative tissue damage aim at the *prevention* of •OH formation, and it may be justified to specifically support these systems, if they are deficient. In short, in a biomedical context we do not need “antioxidants”, neither as scientific term nor as dietary supplements to treat oxidative stress diseases. At best, we can expect a hormetic response from real antioxidants, which often share the tendency to act as pro-oxidants in vivo (see Section 6).

The above examples may suffice to redirect redox biologists to a clear, mostly chemical language. Otherwise, they have to face the risk to end up in a Babylonian confusion of tongues and its consequences, as are described in one of the oldest books in the world (Genesis 11.1-9).

## 8. Conclusions

Redox chemistry can be traced back to the 18th century, while redox biology began in the 19th century with the description of H_2_O_2_ metabolism catalyzed by biological material. Related activities were later attributed to heme proteins such as the heme peroxidases and catalases. Their structural and functional characterization had to wait for the availability of technological progress in synthetic chemistry, spectroscopy, x-ray crystallography, electron spin resonance and stop-flow techniques, and therefore did not reach reasonable maturity before the middle of the 20th century. Although the biological function of catalases in H_2_O_2_ metabolism has meanwhile been established, the role of many heme peroxidases in plants and fungi still awaits clarification.

The second half of the 20th century surprised with two new peroxidase families that proved to lack the catalytic heme moiety. These were the glutathione peroxidases, which work with selenium or sulfur catalysis, and the peroxiredoxins, which overwhelmingly depend on sulfur catalysis. The substrate specificity of these enzymes significantly broadened the scope of redox biology, since these peroxidases proved to also reduce hydroperoxides distinct from hydrogen peroxide. In particular, bio-membrane damage due to lipid peroxidation became a focus of research. Studies on the catalytic mechanism of these enzymes have greatly improved our understanding of the reactivity of protein thiols and selenols.

In the last three decades of the 20th century, redox biology was enriched by two key players that proved to be radicals: the superoxide anion and nitric oxide. The former is built by NADPH oxidases, one of which is a pivotal component of the host defense system, or by the respiratory chain of mitochondria. The superoxide anion is dismutated to H_2_O_2_ and O_2_ by different types of superoxide dismutases. Nitric oxide proved to be the endothelium-derived relaxing factor that is built by the endothelial NO synthase (eNOS). Outside the vascular system, nitric oxide is formed by other NOS-type enzymes and has diverse biological functions.

Up to the middle of the last century, redox biology was dominated by the concern about oxidative damage. By now, the major focus of redox biology is metabolic regulation and signaling by diverse oxygen, nitrogen or sulfur species that proved to be messenger molecules, targets or sensors. The present challenge is to understand how the biological radicals and compounds derived therefrom interact with redox-regulated systems and how the synthesizing and degrading enzymes contribute. A re-interpretation of the biological roles of glutathione peroxidases and peroxiredoxins as possible hydroperoxide sensors and transducers marks the present progress of this research field.

The language of redox biology of the 18th and 19th century is hardly compatible with modern terminology. The language became clearer in the 20th century. More recently, the abundant use of poorly defined terms and abbreviations such as oxidative stress, antioxidants, ROS or RNS has hampered the in-depth understanding of biological redox events. A return to a clearer, chemistry-based language is strongly recommended.

## Figures and Tables

**Figure 1 antioxidants-09-01254-f001:**
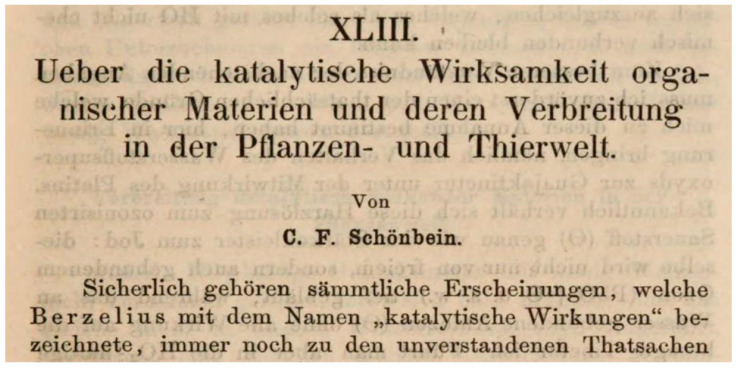
Copy of the title page of Schönbein’s article on the discovery of peroxidases [2]. In the introduction of his article, he characterizes the decomposition of hydrogen peroxide by living materials as a catalytic process and refers to the father of catalysis Jöns Jacob Berzelius (1779–1848).

**Figure 2 antioxidants-09-01254-f002:**
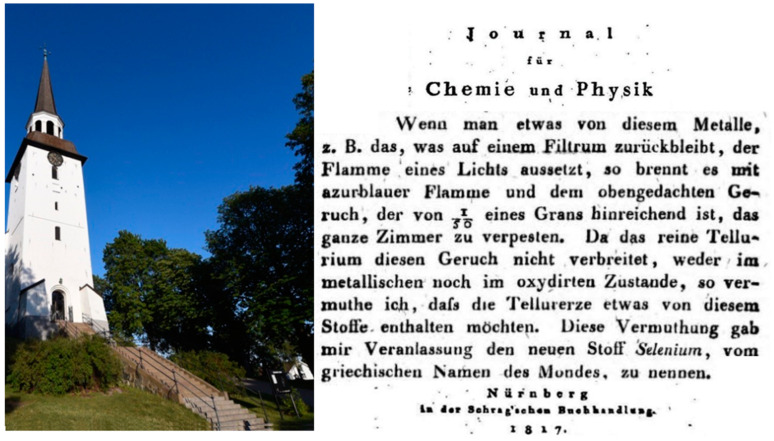
The discovery of selenium. The picture to the left shows Gripsholm village, where Berzelius, in 1817, saw the red mud in the lead chambers of a sulfuric acid factory, which is hidden behind the church. This mud turned out to contain selenium. The little white hut below the trees (right lower corner) was the laboratory of Berzelius (picture taken by L. F. on occasion of the selenium meeting “Se 2017” in Stockholm, organized by Elias Arnér at the Karolinska Institute). The right panel shows an excerpt of the letter of Berzelius to the editor of the *Journal für Chemie und Physik*, J. S. C. Schweigger, in which he reports on the discovery of selenium for the first time. The letter is dated January 27, 1818, but published in a volume of 1817 [59].

**Figure 3 antioxidants-09-01254-f003:**
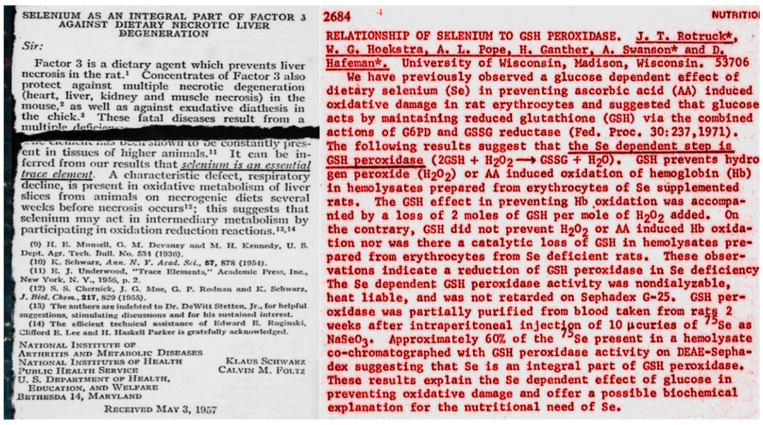
Milestones of biochemical selenium research. In the left panel the essentiality of selenium in mammals is documented for the first time [64]. Please, notice that the co-inventor, Dewitt Stetten, who smelled the selenium in the “factor 3” of Klaus Schwarz, is hidden in footnote 13. The right panel, colored in selenium red, claims, for the first time, that the GPx reaction depends on selenium [65]. It is the communication that prompted us to try an exact selenium determination in the last 0.69 mg of bovine GPx1 [66] left over from material-consuming kinetic studies with stopped-flow equipment [67].

**Figure 4 antioxidants-09-01254-f004:**
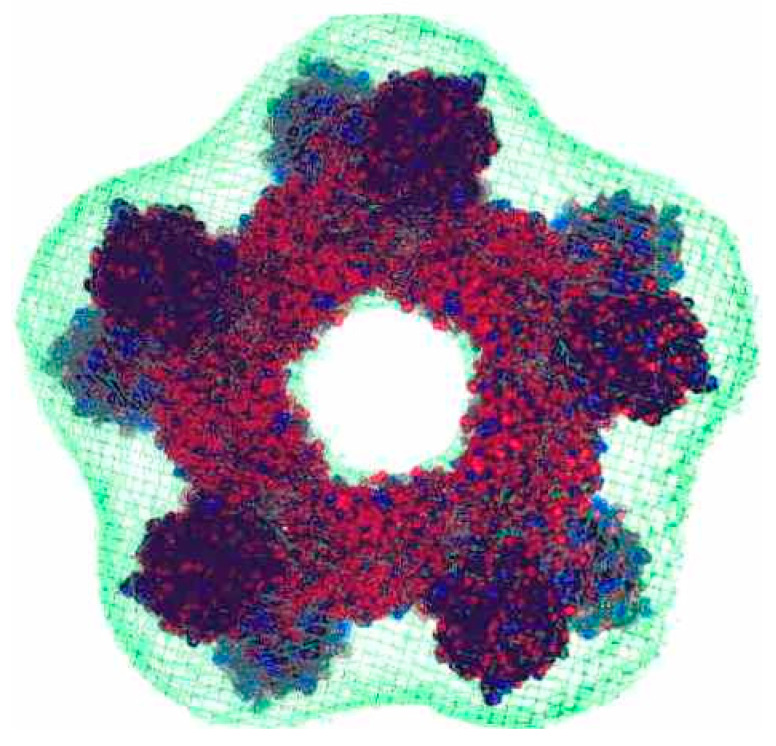
Model of tryparedoxin peroxidase of *Trypanosoma brucei brucei* loaded with 10 molecules of mutated tryparedoxin. Tryparedoxin peroxidase is a typical 2-Cys-Prx that tends to form donut-shaped decamers consisting of five catalytic units (dimers). Here, the oxidized tryparedoxin peroxidase was reacted with an excess of a tryparedoxin mutant with the C-terminal cysteine of the CPPC motif changed to serine (*Tb*TXNC43S). The surface-exposed C40 of the tryparedoxin mutant can still react with the resolving cysteine (C173´) of the peroxidase, but stays attached (protrusions at the donut), because it can no longer fully reduce the peroxidase [139]. The model is based on electron microscopic images as described in detail in [140] and was prepared by H. J. Hecht.

**Figure 5 antioxidants-09-01254-f005:**
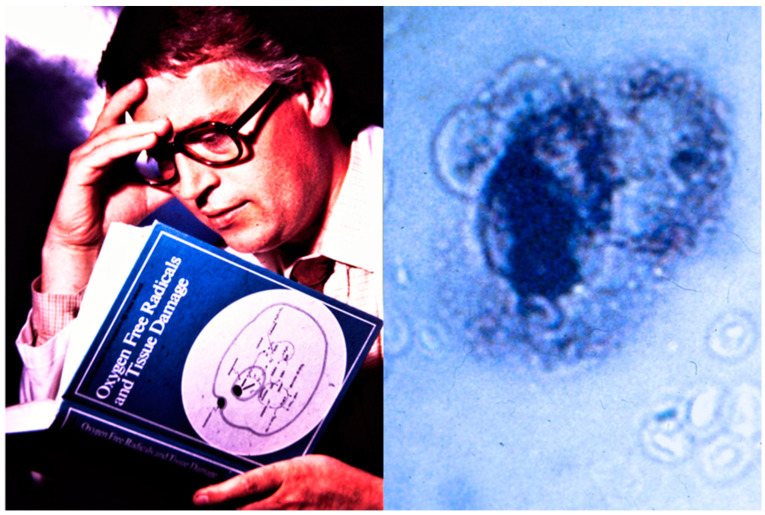
Gerriet Loschen reading the proceedings of the Ciba Symposium on “Oxygen Free Radicals and Tissue Damage” [239]. The cover illustration schematically represents a phagocytosing white blood cell, as shown in [240]. The picture was taken in 1980 by L. F. The right panel shows a more realistic version of phagocytosis: a white blood cell swallowing opsonized yeast (phase contrast microscopy). •O_2_^−^ is stained with tetrazolium blue. Notice that most of the dye is precipitated in the phagocytic vacuole, but also faintly spread over the entire surface of the cell. The right picture was taken by G. L. in 1980.

**Figure 6 antioxidants-09-01254-f006:**
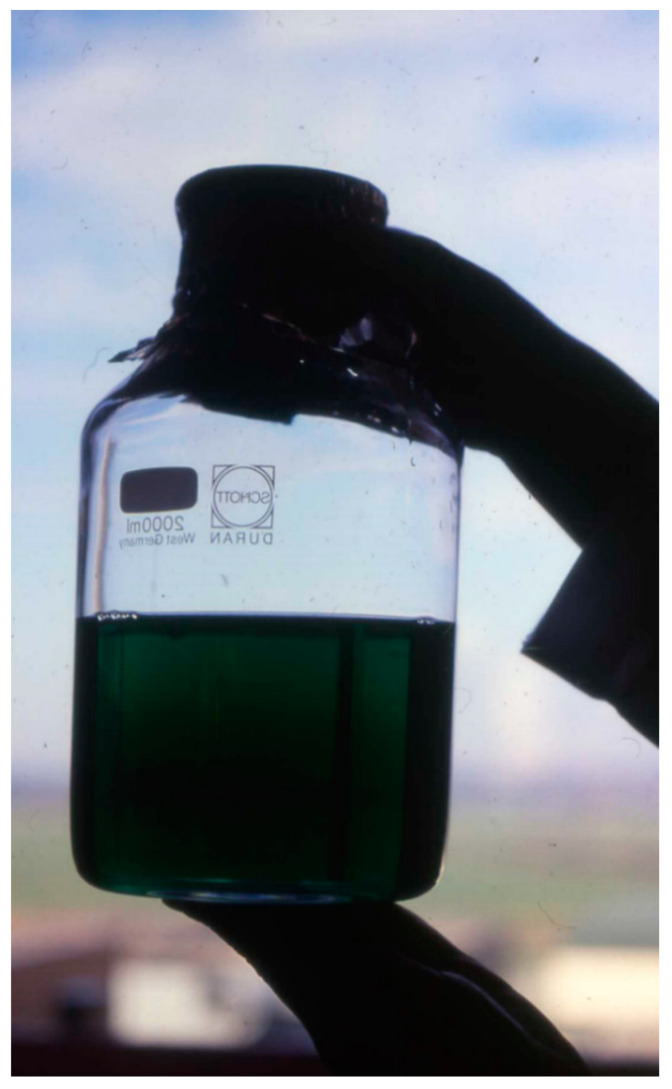
First bottle of human Cu/Zn-SOD isolated from genetically modified *Escherichia coli*. The bottle contains the whole yield of a 100-l-fermenter; the green solution looks almost black. The poor picture (taken by L. F.) was proudly shown in 1985 at the symposium on “Superoxide and Superoxide Dismutases in Chemistry, Biology and Medicine” organized by Guiseppe Rotilio in Rome (Italy) [242].

**Figure 7 antioxidants-09-01254-f007:**
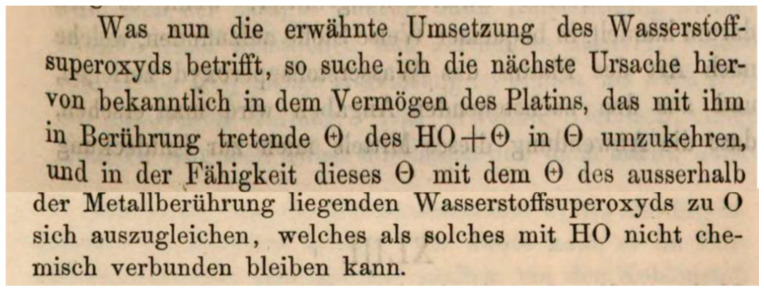
Chemical speculations of the 19th century. Excerpt of the article by Schönbein [2] that tries to explain the catalytic mechanism of peroxidases as analogous to noble metal-induced decomposition of hydrogen peroxide. Language and contents are not easily understood: H_2_O_2_ is called “Wasserstoffsuperoxyd”, a name now reserved for •O_2_H; the presumed composition of hydrogen peroxide (O_2_H) is simply wrong; the analogy of the peroxidase reaction and metal-induced peroxide decomposition may be doubted; the characterization of oxygen atoms of differing degrees of reactivity is “unique”.

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
