# Peer review of "Looking Back at the Early Stages of Redox Biology"

_antioxidants, 2020, doi:10.3390/antiox9121254_

Round 1

Reviewer 1 Report

This is a fascinating journey through the history of redox biochemistry. I really enjoyed it. I do have a couple of suggestions and a very minor point, that there are several typographical errors. Comments:

  • I know this is mainly about animals, but it would be useful to cite recent papers on nitric oxide history and terminology, and that the notion of a redox zone has been mooted, which would be good to mention.
  • A few typos I spotted:

Succesfully spelt wrong.

Function should be functions – near [26].

“In kind of introductory chapter “ is an odd phrase.

There were a few others, but with the lack of line numbers I stopped listing them.

Author Response

The author wishes to thank the referee for his overall positive statement.

  1. Some topical reviews have been added at the end of Section 4.2.

Reviewer 2 Report

In this review manuscript, the author provides a historical perspective on how the field of ‘redox biology’ has developed.  The manuscript offers an informative, insightful reflection on major developments in the field over the past two centuries.  I thoroughly enjoyed reading it and feel that it will be a useful resource for new (and established) scientists in the field.  Most of my comments are ‘line edits’ that are intended to add clarity to the wording in particular passages or that correct typographical errors that I noticed, although I do have a handful of additions that the author may consider when revising the manuscript.  I recommend accepting this manuscript for publication once these minor revisions are complete. 

Scientific considerations in a revised manuscript

  • On page 4, when discussing how the biological function for the majority of peroxidases remains obscure in the second paragraph, it may be useful to point out whether genetic studies with knockout organisms have yielded any insights into whether the heme peroxidases are truly protective against peroxides and/or other oxidants. I am uncertain whether these experiments have been performed, but they could suggest a role for the enzymes in oxidant defense in addition to the other biological processes described.

  • In the selenium section (at the bottom of page 6), it could be worth foreshadowing that not all GPx enzymes–especially those in microbial species–are selenoproteins.This observation comes up later, but it might be worth hinting at here.

  • In Fig. 4, the tryparedoxin peroxidase seems to be a specialized example of a typical 2-Cys Prx, given that it does not form the typical “doughnut-shaped” decamer observed with other enzymes in this class. I found myself asking what the extra appendages hanging off of the decamer are. It may be useful to compare to other Prx structures like that of Prx1 or Prx2 from mammals or yeast Tsa1, as these are, arguably, the prototypes for typical 2-Cys Prx structures. 

  • On page 11, in the paragraph beginning “Certainly, the examples…” – it might be worth calling the redox signaling described here by the common terminology in the field, noting the role thiol-based “redox switches” play in transducing redox signals and describing these mechanisms in a bit more chemical detail.

  • On page 21, in the paragraph beginning “The known molecular targets…,” perhaps it is worth noting that there have been recent advances in understanding methionine oxidation and its role in regulating protein activity.The paragraph mainly focuses on cysteine thiol oxidation (where, understandably, there is more known), but I do think there have been inroads in the area of methionine thioether oxidation and its role in altering biological processes in the past five years.

Typographical errors/suggested phrasing changes…

Page 1

Abstract

-change ‘nitrogen monoxide radical’ to ‘nitric oxide,’ since ‘nitric oxide’ is more common

Introduction – first paragraph

-‘synthetized’ should be ‘synthesized’

-‘consumption of energy derived from redox processes’ is thermodynamically impossible; ‘consumption of ATP derived from redox processes’ seems more accurate

Page 3 – second full paragraph

-‘the formation of compound I was’ rather than ‘were’

-‘pigeon’ rather than ‘pigean’

-‘successfully’ rather than ‘succesfully’

Page 4 and elsewhere

-it may be helpful to refer to ‘Section’ numbers rather than ‘Chapter’ numbers

Page 4, last line

-‘photocopying’ rather than ‘fotocopying’

Page 6, figure legend

-‘co-inventor’ rather than ‘co-inventors’

Page 9, first full paragraph

-include a period at the end of the last sentence

Page 9, third full paragraph

-I think ‘TSH’ should be ‘TSA,’ and ‘homologous’ is misspelled

-point out the kingdom for Crithidia fasciculata for those of us who are unfamiliar with this organism

-‘In the kinetoplasts’ should be followed by a comma rather than a period

Page 11, first full paragraph

-unclear what is meant by ‘a critical cysteine dissociates.’ Do you mean that ‘the proton from a critical cysteine thiol dissociates to form a thiolate’?

Page 11, second full paragraph

-‘scarce’ rather than ‘scares’

Page 11, final paragraph

-might be worth pointing out ‘Tobias Dick’s group’ rather than ‘Sarah Stocker and colleagues’

Page 12, second paragraph

-‘to eliminate the 2’-OH group of ribose in ribonucleotides’ might be a simpler and more accurate way of stating

Page 12, fourth paragraph

-‘conjugate’ acid rather than ‘conjugated’ acid

Page 13, first full paragraph

-‘zinc’ rather than ‘zink’

Page 13, second full paragraph

-there’s an extra space after reference ‘[182] .’

Page 15, first full paragraph

-‘as a possible source’ rather than ‘as possible source’

Page 15, second full paragraph, near the end

-‘ubisemiquinone’ is misspelled

Page 17, first paragraph

-‘In animal experiments,’ rather than ‘In animal experiment,’

Page 17, last line

-there’s an extra space between the open parentheses and p91phox

Page 18, first full paragraph

-‘The source is’ rather than ‘are’

Page 18, last paragraph

-‘guanylyl cyclase’ rather than ‘guanolyl cyclase’

-‘In mammals, •NO biosyntehsis’ rather than ‘its biosynthesis’ would add clarity

Page 19, first bullet point

-‘worked out before’ rather than ‘worked out, before’

Page 19, second bullet point

-for sake of consistency, ‘guanylyl cyclase’ rather than ‘guanosylyl cyclase’

Page 20, first bullet point

-‘peroxynitrite’ is misspelled

Page 21, second full paragraph

-omit ‘the’ in front of ‘the ferroptosis’

Page 22, first full paragraph

-‘blocks’ rather than ‘block’

-there’s a space between the open parentheses and disheveled

Author Response

The author thanks for the referee 2 for the overall positive statements and his long list of typos, grammar mistakes and rephrasing suggestions.

  1. The typos, mistakes and rephrasing suggestions were all considered as requested.

  1. Genetic studies on plant peroxidases are scarce. As far as available, the data support the view that class III peroxidases are not involved in detoxification of H2O2, while those showing catalase activity and ascorbate peroxidase are typical antioxidant enzymes. Supporting references have been added to the paragraph.

  1. CysGPxs have now been introduced already in section 3 with reference to section 4.1.

  1. The legend of Fig. 4 has been extended to explain how the model of the substrate-loaded tryparedoxin peroxidase was obtained (including new reference).

  1. The term “redox switch” has now been introduced in § 3 of section 4.1.

  1. Regulatory methionine oxidation has now been mention and referenced in the last but one paragraph of section 6.

Other changes

  1. After publication of the preprint I received some critical comments from experts in the field (H. Sies, University of Düsseldorf, and W. Koppenol, ETH Zürich). H. Sies found my critique of the term “Oxidative Stress” a bit too negative. W. Koppenol inter alia mailed me two PNAS articles saying that singlet oxygen is not formed during the decomposition of peroxynitrite. He also insisted that singlet oxygen is not formed directly by the myeloperoxidase reaction. I made appropriate corrections or amendments and rephrased part of section 7.
  2. I added a conclusion section.
  3. The references were produced using EndNote and the MDPI style. The outcome was a disaster: Titles of articles were throughout printed in lower case (also German titles, abbreviations such as ATP, NADPH, GAPD or element symbols (Cu or Zn) and the like. These mistakes have been corrected (I hope that I found them all). Moreover, journal names were sometimes abbreviated, sometimes not; abbreviations were sometimes marked with a full stop, mostly not; journal names were mostly printed in mixed upper case/lower case, sometimes not etc. etc. I tried to make the reference list consistent as follows:
  • I removed the full stops after abbreviations.
  • Journal names have now been printed as usual in a mixed upper case/ lower case mode and in italics (e. g. J Biol Chem).
  • Journal names which were printed in full by EndNote were not abbreviated, because this way they provide optimum information and I do not know what kind of abbreviation is preferred by Antioxidants.
  • Book titles have been treated like journal names ( e. g. The Enzymes)

A Manuscript with all changes of the main text marked in red is attached.

I look forward to your final decision.

Sincerely yours

Leopold Flohé